

# Motor expertise affects the unconscious processing of geometric forms

Jiaxian Geng[1,*], Fanying Meng[1,2,*], Chao Wang[3], Hanna Haponenko[4] and Anmin Li[2]

[1] Institute of Physical Education, Huzhou University, Huzhou, Zhejiang, China
[2] School of Psychology, Shanghai University of Sport, Shanghai, China
[3] Department of Psychology, School of Teacher Education, Huzhou University, Huzhou, Zhejiang, China
[4] Department of Psychology, Neuroscience & Behaviour, McMaster University, Hamilton, Ontario, Canada
[*] These authors contributed equally to this work.

## ABSTRACT

**Background**. The unconscious processing of information is an important skill used by competitive athletes to handle the rapidly changing movements of opponents and equipment. Previous studies have shown that unconscious information processing among athletes is better than that among non-athletes in the sports-specific domain. However, it is not yet clear whether athletes also show superior unconscious information processing in the general cognitive domain.

**Methods**. Twenty-five competitive table tennis players (athletes) and 26 aged-matched non-athletic college students (non-athlete controls) were recruited for this study. Participants first performed a masked priming task that used geometric shapes as primes and targets to examine unconscious information processing in the general cognitive domain. As a control, participants then completed a prime identification task to determine whether they could consciously detect the priming geometric forms. Reaction times and error rates were analyzed to examine whether motor expertise influenced unconscious information processing in the general domain. Nineteen athletes and 17 non-athletes from our present study, which used general stimuli, also participated in our previous study, which used sport-specific stimuli. The strength of the unconscious response priming effect was analyzed to examine whether the effect of motor expertise on unconscious processing could be transferred from a sports-specific domain to a general domain.

**Results**. Signal detection analyses indicated that neither athletes nor non-athletes could consciously perceive the priming stimuli. Two-way repeated-measures analyses of variance followed by simple main effects analyses of the masked priming performance, indicating that athletes responded faster and committed fewer errors when the priming stimulus was congruent with the target stimulus. These results suggested that athletes exhibited a significant unconscious response priming effect of geometric forms. By contrast, non-athletes did not respond faster or commit fewer errors for congruent vs. incongruent conditions. No significant difference was detected between athletes and non-athletes in error rates for congruent trials, but athletes committed significantly more errors than non-athletes on incongruent trials. The strength of the unconscious response priming effect that athletes exhibited was greater than that for non-athletes, both in the present study with general stimuli and in our previous study with sport-specific stimuli.

Corresponding author
Anmin Li, anminli@sus.edu.cn

**Conclusion**. The results indicated that motor expertise facilitated the unconscious processing of geometric forms, suggesting that the influence of motor expertise on unconscious information processing occurs not only for the sports-specific domain but also transfers to the general cognitive domain.

# INTRODUCTION

Previous studies have shown that behavior can be influenced by visual stimuli that are not consciously perceived; individuals can encode information that lies below their threshold for conscious awareness (*Ortells et al., 2016*; *Tseng et al., 2016*; *Zovko & Kiefer, 2013*). In competitive sports, athletes, especially those engaged in sports that require open skills, frequently process movement information unconsciously because of the limited time available prior to motor reaction (*Kibele, 2006*). A widely used paradigm to study unconscious information processing is the masked priming task (*Kiefer, 2012*; *Kiefer & Martens, 2010*). The unconscious response priming effect is considered an indirect measure in evaluating the processing of masked primes (*Ansorge, Kunde & Kiefer, 2014*). The theory of direct parameter specification (DPS, *Neumann, 1990*; *Neumann & Klotz, 1994*) could be used to explain the unconscious response priming effect. According to the DPS theory, once an action plan is established, sensory information can specify response parameters without conscious perception. In the case of the masked priming paradigm, researchers have proved the specification of response parameters by the masked prime through EEG (*Leuthold & Kopp, 1998*; *Minelli, Marzi & Girelli, 2007*) and fMRI modeling (*Dehaene et al., 1998*; *Dostilio & Garraux, 2011*; *Ulrich & Kiefer, 2016*).

Previous studies have suggested that long periods of practice are required for unconscious or implicit learning (*Lewicki, Czyzewska & Hoffman, 1987*; *Wulf & Schmidt, 1997*). With the masked priming paradigm, recent studies found that athletes exhibit stronger unconscious response priming effects than non-athletes in a sport-specific domain, suggesting that athletes show superior unconscious information processing owing to their extensive specialized training (*Gueldenpenning et al., 2015*; *Meng et al., 2019*). However, whether athletes also show superior unconscious information processing in the general domain is still unknown. Expertise or practice may be an important prerequisite for unconscious information processing (*Heinemann et al., 2010*; *Kiesel et al., 2009*; *Reuss et al., 2015*). Athletes are considered experienced individuals because of the extensive motor expertise they have accumulated during their years of training (*Guo, Li & Yu, 2017*; *Voss et al., 2010*). Additionally, unconscious information processing is an important skill that enables athletes to deal with rapidly changing movement information (*Gueldenpenning et al., 2011*; *Kibele, 2006*). Hence, researchers are beginning to investigate the relationship between motor expertise and unconscious information processing. For example, using the masked priming task, *Meng et al. (2019)* investigated unconscious information processing among

table tennis players in the context of objects that represented table tennis—notched circles. The notched circle stimuli were designed based on the characteristic of table tennis and a theoretical model of long-time object memory, called the type-token model. The researchers believed that the circle provided outline information, and the notch orientation provided directional information of the hitting point for the table tennis (*Guo et al., 2017*). Results showed that table tennis players, compared to a non-athletic control group, unconsciously perceived visuospatial information and showed a significant unconscious response priming effect. Using a similar methodology, *Gueldenpenning et al. (2015)* also examined the unconscious processing of complex movements. They researchers found that athletes involved in martial arts training, compared to novices, were able to unconsciously distinguish feint and non-feint actions and to initiate faster motor responses. Some researchers have suggested that the high perceptual sensitivity and the perceptual-motor common representations among athletes may explain their superiority in unconscious information processing (*Gueldenpenning et al., 2011*). However, the stimuli that were used in these aforementioned studies are associated with the sports-specific domain. Whether the effect of motor expertise can be transferred from a sports-specific domain to a general domain has not been investigated.

The present study examined whether motor expertise facilitates unconscious information processing in a general masked priming task that uses stimuli unrelated to sports. We recruited a group of competitive table tennis players and a group of aged-matched college students without specific motor expertise to perform a masked priming task and a control prime visibility task. We hypothesized that both athletes and non-athletes would respond faster and commit fewer errors on trials in which prime and target stimuli were congruent than when they were incongruent, displaying a significant unconscious response priming effect. We further hypothesized that the strength of this unconscious response priming effect in athletes would be stronger than that in non-athletes.

## MATERIALS AND METHODS

### Participants

To find an interaction between expertise and response congruency in line with *Meng et al. (2019)*, we calculated the sample size by Gpower3.1 ($\alpha = 0.05$, power $= 0.80$, effect size $= 0.19$). The result showed that a minimum of 40 participants was required. Twenty-five aged-matched competitive table tennis players (9 females; mean age, 20.44 years with a range of 18–23 years) and 26 college students (10 females; mean age, 19.85 years with a range of 18–22 years) were recruited for this study. The competitive table tennis players were considered athletes, and the college students were considered non-athlete controls. All athletes were recruited from Chinese Table Tennis College, Shanghai University of Sport. All of them had obtained the first or second level of the national standard, with a mean table tennis training experience of 7.96 years. The non-athletes were recruited from the School of Economics and Management, Shanghai University of Sport, and had no practical experience with any sports. All participants were right-handed, had normal or corrected-to-normal vision, and had no psychiatric or neurologic illness. All participants

provided written informed consent before starting the experiment and received financial compensation after finishing the experiment. This study received approval from the Ethics Committee of Shanghai University of Sport (No. 2018025).

## Stimuli and equipment

On the basis of previous studies (*Ulrich & Kiefer, 2016*; *Zovko & Kiefer, 2013*), we selected images of four black geometric forms (circle, diamond, ellipsoid, and square) as primes and targets. Two images that contained many randomly oriented lines were designed as masks. The stimuli (including masks) were presented on a gray background (RGB values of 128, 128, and 128) with a size of 7 × 7 cm and subtended a visual angle of 6.69° (horizontally and vertically) from a viewing distance of 60 cm.

A Dell computer with a 16-inch VGA display (frequency 60 Hz, resolution 1,024 × 768) was used for stimulus presentation. The E-prime 2.0 software package (Psychology Software Tools, Pittsburgh, PA, USA) was used for response sampling.

## Procedure and tasks

### Masked priming task

During the experiment, the experimenter strictly adhered to the rules and maintained a neutral attitude. Participants were instructed to respond to the geometric form of the target stimuli as soon as possible under the premise of ensuring the correct response by pressing the appropriately assigned key on a computer keyboard. An image of a circle or a diamond was assigned to the "f" key, and an image of an ellipsoid or a square was assigned to the "j" key. The reaction was counterbalanced across participants; half the participants followed the aforementioned stimulus-response key assignment, and the remaining participants followed the opposite stimulus-response key assignment.

The sequence of each trial in the masked priming task is shown in Fig. 1 and occurred as follows: (1) a fixation cross appeared for 750 ms; (2) a forward mask (i.e., an image containing many randomly oriented lines) appeared for 200 ms; (3) the priming stimulus appeared for 33 ms; (4) a backward mask (i.e., another image containing many randomly oriented lines that differed from the forward mask) appeared for 33 ms; (5) the target stimulus appeared for 500 ms; (6) a blank screen appeared for 1,000 ms. Participants were required to report their decision (i.e., press the appropriate key) within 1,500 ms. The inter-trial interval varied randomly from 1,000 to 1,500 ms.

All participants were given 24 practice trials before the formal experiment began. The masked priming task followed and included four blocks of 168 total trials. The relationship between the prime and target allowed for two experimental conditions: congruent, for which the prime and target included geometric forms belonging to the same response category, and incongruent, for which the prime and target included geometric forms belonging to the different response category. Within each block, half the trials were congruent, and half were incongruent. The four geometric forms appeared equally often as primes and targets and were varied across trials to avoid the repetition response priming effect.

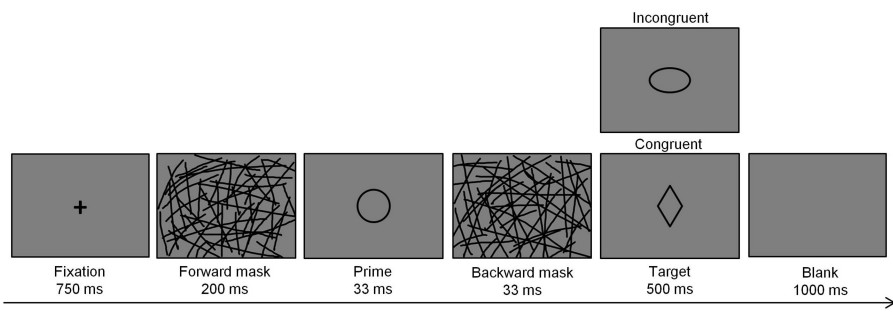

**Figure 1** **The masked priming task.** Trial sequence of the masked priming task.

### Prime identification control task

To assess whether or not participants consciously perceived the prime, they participated in a task that provided a subjective measure of conscious awareness after finishing the masked priming task. This measure was conducted in an interview. The sequence and content of questions occurred as follows: (1) In addition to the fixation, two random line patterns, and target graphics, what else did you see? (2) Did you see any flashing stimulus between the two random line patterns? (3) What do you think this flashing stimulus was? (4) Did you see a geometric form during this flashing stimulus? (5) What specific geometric form did you see? In addition, the participants performed an objective measure of prime identification. Apart from 24 practice trials, the prime identification task included 64 trials (32 congruent trials and 32 incongruent trials). Participants were informed of the existence of the prime. The prime identification task procedure was identical to the masked priming task in that participants were asked to respond to the geometric form of the primes with the same stimulus-response key assignment as was used for the priming task. There was no time pressure for participants in the prime identification task. Participants were required to make their best guess when they were unable to determine the form of the primes.

## Statistical analysis
### Identification rates

We used the statistic $d'$ as a signal detection measure to assess prime visibility. Consistent with the work of *Green & Swets (1966)*, the present study considered a correct response to the target stimulus on a congruent trial a hit and an erroneous response to the target stimulus on an incongruent trial a false alarm (*Kiefer & Martens, 2010*). The hit rates and false-alarm rates were calculated for each participant, and the identification rate $d'$ was calculated from each participant's hit and false-alarm rates.

To obtain an objective measure of the ability to identify the geometric prime form presented between the two masks, one-sample $t$-tests were performed to determine whether the identification performances of athletes and non-athletes were distributed around the level of chance, at 50% (which is indicated by $d' = 0$). We then compared the identification rates of priming stimuli ($d'$) between athletes and non-athletes through independent $t$-test. The Bayes Factor was employed to evaluate the null-hypothesis (a Cauchy distribution (scale = 0.707) as the prior for the one-sample $t$-test). To rule out
that any unconscious response priming effect observed among athletes correlated with prime visibility, Pearson's correlation between the identification rate $d'$ and the size of the unconscious response priming effect was performed for the group of athletes.

### Reaction times and response errors

To minimize errors caused by incorrect and extreme values, incorrect and missed trials (17.52%) and reaction times (RTs) that deviated by more or less than three standard deviations (0.83%) were removed from further analysis. The mean RT of the correct responses and the mean error rates (ERs) were calculated for each participant and each experimental condition. Two-way repeated-measures analyses of variance (ANOVAs) were used to examine the mean RTs and ERs, with the within-subjects factor being response congruency (congruent vs. incongruent) and the between-subjects factor being expertise (athletes vs. non-athletes).

### The strength of the unconscious response priming effect

To better illustrate whether the effect of motor expertise on unconscious processing could be transferred from a sports-specific domain to a general domain, we first selected the 19 athletes (6 females; mean age, 20.63 years with a range of 18–23 years) and 17 non-athletes (5 females; mean age, 20.06 years with a range of 18–22 years), these subjects participated in the present study with general stimuli, as well as the previous study with sport-specific stimuli. The strength of the unconscious response priming effect (the reaction time on incongruent trials minus the reaction time on congruent trials) was calculated for each participant and each experimental condition. Two-way repeated-measures analyses of variance (ANOVAs) were used to examine the strength of the unconscious response priming effect, with the within-subject factor being stimulation type (general stimuli vs. sport-specific stimuli) and the between-subjects factor being expertise (athletes vs. non-athletes).

## RESULTS

### Prime visibility

During the subjective measurement, 10 athletes and 7 non-athletes reported that they saw some images between the random line patterns, but the exact content of the images was unknown. Additionally, four athletes and two non-athletes reported that they saw some regular forms, but it was not clear what the forms were. The other participants did not perceive any images except the fixation, random lines patterns, and target graphics.

The signal detection analysis on the performance of participants to identify the masked primes indicated that, for athletes, $d'$ was 0.14, which did not deviate significantly from zero ($t_{(24)} = 0.97$, $p = 0.341$). In addition, for non-athletes, $d'$ was 0.06, which also did not differ significantly from zero ($t_{(25)} = 0.81$, $p = 0.425$). Furthermore, no significant difference was found between the $d'$ values of athletes and non-athletes ($t_{(49)} = 0.48$, $p = 0.636$). These results indicated that neither athletes nor non-athletes could consciously perceive the masked primes.

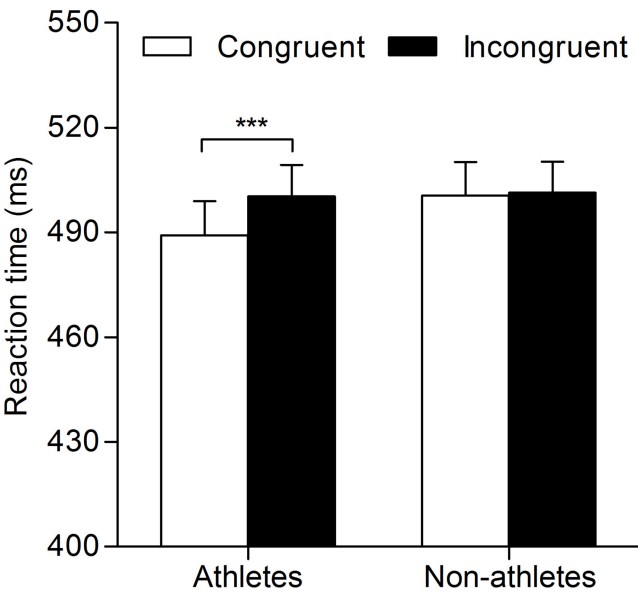

**Figure 2** **Mean reaction times for trials with correct responses in the masked priming task for athletes and non-athletes.** The vertical bars represent mean reaction times as a function of response congruency and expertise. White bars represent reaction times of congruent responses, and black bars represent reaction times of incongruent responses. Error bars indicate standard error of mean; ***$p < 0.001$ for the indicated comparison.

The distribution of the identification rates $d'$ (Kolmogorov-Smirnov = 0.14, $p = 0.20$) and the size of the unconscious response priming effect (Kolmogorov-Smirnov = 0.10, $p = 0.20$) for athletes were normal. For athletes, the identification rates $d'$ and the size of the unconscious response priming effect did not correlate with each other ($r(25) = -0.19$, $p = 0.377$), suggesting that the unconscious response priming effect of the athletes was not the result of their awareness of the masked primes.

### Reaction times and response errors

The results of a two-way repeated-measures ANOVA examining RTs revealed a significant main effect of response congruency ($F_{(1,49)} = 11.44$; $p = 0.001$; $\eta_p^2 = 0.19$). The interaction between expertise and response congruency was also significant ($F_{(1,49)} = 8.43$; $p = 0.006$; $\eta_p^2 = 0.15$). A simple effects analysis of the interaction showed that athletes responded faster on congruent trials than on incongruent trials (mean ± standard error: congruent, 489.15 ± 9.75 ms; incongruent, 500.33 ± 8.99 ms; $p = 0.00$). By contrast, there was no significant difference in reaction times between congruent and incongruent trials for non-athletes (congruent, 500.56 ± 9.56 ms; incongruent, 501.41 ± 8.82 ms; $p = 0.734$). In addition, the main effect of expertise did not reach statistical significance ($F_{(1,49)} = 0.23$; $p = 0.633$; $\eta_p^2 = 0.01$) (Fig. 2).

An analogous two-way repeated-measures ANOVA on ERs revealed a significant main effect of response congruency ($F_{(1,49)} = 38.39$; $p = 0.00$; $\eta_p^2 = 0.44$). The interaction between expertise and response congruency was also significant ($F_{(1,49)} = 22.80$; $p = 0.00$; $\eta_p^2 = 0.32$). For response congruency, the analysis of the simple effects of the interaction

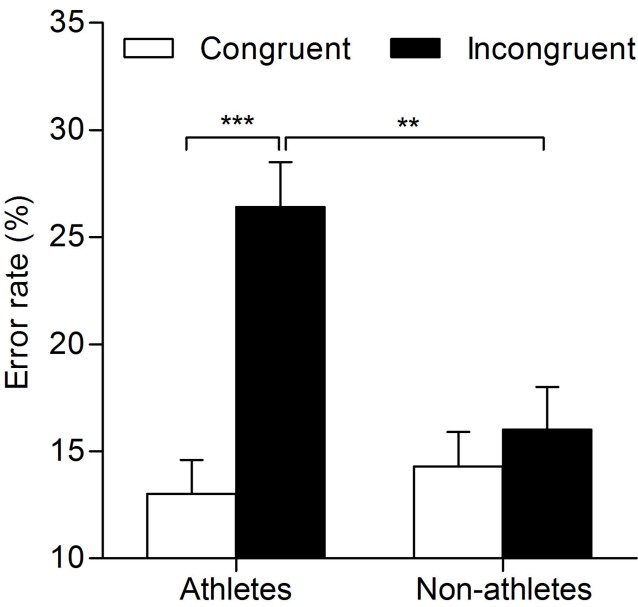

**Figure 3** **Mean error rates for athletes and non-athletes in the masked priming task for athletes and non-athletes.** The vertical bars represent mean error rates as a function of response congruency and expertise. White bars represent error rates of congruent responses, and black bars represent error rates of incongruent responses. Error bars indicate standard error of mean; **$p < 0.01$, ***$p < 0.001$ for the indicated comparisons.

showed that athletes committed fewer errors on congruent trials than on incongruent trials (congruent, 13.00% ± 1.59%; incongruent, 26.36% ± 2.08%; $p = 0.001$). By contrast, there was no significance difference in ERs between congruent and incongruent trials for non-athletes (congruent, 14.27% ± 1.56%; incongruent, 16.00% ± 2.04%; $p = 0.572$). For expertise, athletes committed more errors than non-athletes on incongruent trials (athletes, 26.36% ± 2.08%; non-athletes, 16.00% ± 2.04%; $p = 0.00$), but not on congruent trials (athletes, 13.00% ± 1.59%; non-athletes, 14.27% ± 1.56%%; $p = 0.315$). However, the main effect of expertise did not reach significance ($F_{(1,49)} = 3.94$; $p = 0.053$; $\eta_p^2 = 0.07$) (Fig. 3).

### The strength of the unconscious response priming effect

The results of a two-way repeated-measures ANOVA examining the strength of the unconscious response priming effect revealed a significant main effect of expertise ($F_{(1,34)} = 12.58$; $p = 0.001$; $\eta_p^2 = 0.27$). The strength of the unconscious response priming effect that athletes showed was higher than for non-athletes (athletes, 15.13 ± 2.65 ms; non-athletes, 1.46 ± 2.80 ms). The main effect of stimulation type did not reach statistical significance ($F_{(1,34)} = 0.05$; $p = 0.818$; $\eta_p^2 = 0.00$). The interaction between expertise and stimulation type did not reach statistical significance ($F_{(1,34)} = 1.51$; $p = 0.228$; $\eta_p^2 = 0.04$) (Fig. 4).

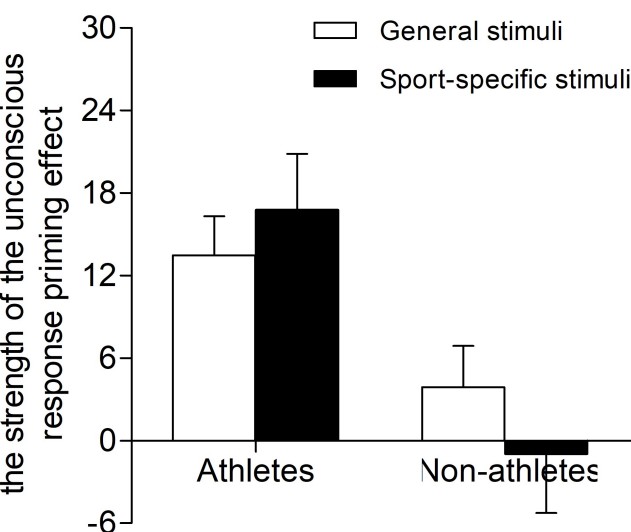

**Figure 4** **Mean strength of unconscious response priming effect in the masked priming task with general stimuli and sport-specific stimuli for athletes and non-athletes.** The vertical bars represent mean strength of unconscious response priming effect as a function of stimulation type and expertise. White bars represent the strength of unconscious response priming effect of general stimuli, and black bars represent the strength of unconscious response priming effect of sport-specific stimuli.

## DISCUSSION

Although unconscious information processing among athletes is reportedly better than that of non-athletes in the sports-specific domain, previous research has not fully shown whether athletes also show superior unconscious information processing in the general domain. Thus, to determine whether motor expertise is associated with unconscious information processing in the general domain, our experiment compared the performance between athletes and non-athletes on a masked priming task that used geometric forms unrelated to the sport domain as the prime and target. The athletes belonged to a group of competitive table tennis players who had extensive motor expertise, and the non-athletes belonged to a group of college students without specific motor expertise. Our primary results showed that athletes responded faster and committed fewer errors on trials in which the prime and target stimuli were congruent (i.e., required the same response) than on trials in which they were incongruent. By contrast, there was no significant difference in reaction times or error rates between congruent and incongruent trials for non-athletes. Moreover, athletes showed a significant unconscious response priming effect both in the masked priming task with general stimuli and the masked priming task with sport-specific stimuli.

Athletes showed a significant unconscious response priming effect compared to non-athletes, which suggests that athletes exhibited superior unconscious information processing. This finding is consistent with the results of a previous study showing that motor expertise promotes unconscious response inhibition in the general cognitive domain (*You et al., 2018*). We speculate that motor expertise possessed by athletes

facilitates unconscious information processing. A characteristic of competitive sports is that they require rapid and changeable responses. Given the Olympic motto "faster, higher, and stronger" (*Coubertin, 2008*), competitive sports, especially open skills sports, have developed so that the movement speed of athletes and sports equipment exceeds the perception threshold of individuals most of the time. For example, in table tennis, the ball speeds may be up to 144–180 km/h, and the contact time between the ball and the racket is only approximately 0.01 s (*Bootsma & Van Wieringen, 1990*). With such time pressure, the decisions made by competitive athletes are usually without any explicit evaluation of the perceived information (*Raab & Johnson, 2008*; *Williams & Ward, 2007*). Table tennis is an open skills sport in which athletes are required to react in a dynamically changing environment, and likely because of their years of training, these athletes show more unconscious behaviors compared with non-athletes (*Wang et al., 2013*; *Yu et al., 2017*). We further found that the strength of the unconscious response priming effect among athletes was not affected by stimulation type, suggesting that the influence of motor expertise on unconscious processing can be transferred from a sports-specific domain to a general domain. The masked priming paradigm used in the present study closely replicated the time pressure of an open skills sport and was favorable for the performance of the athletes.

*Schütz-Bosbach & Prinz (2007)* have suggested that perceptual sensitivity is a prerequisite for unconscious processing. We selected geometric forms as prime and target stimuli because they are considered well-known to most people (*Kiesel et al., 2009*). Thus, we believed that both athletes and non-athletes would show high perceptual sensitivity to these geometric forms. However, our results showed that only athletes exhibited a significant unconscious response priming effect whereas the non-athletes did not. This result was in line with *Guo et al. (2017)*, which found that regardless of priming with sport-specific stimuli or general stimuli, table tennis players had better performance in perceptual observation and motion control compared to non-athletes. According to the results of the present study and previous research, we speculated that athletes had higher perceptual sensitivity to geometric forms in comparison with non-athletes. To accurately judge the flight trajectory of the ball, as well as its spinning and land point, table tennis players may rely heavily on the visual imagery system. Athletes may process movement information through image coding during daily training and competition. Therefore, it may be the case that athletes with extensive special training showed higher perceptual sensitivity to the sport-specific stimuli and general geometric forms compared to non-athletes.

In fast-ball sports, the perception-action coupling capacities may play a critical role in motor expertise (*Ranganathan & Carlton, 2007*). These capacities are not limited to sport-specific scenarios, but may also differentiate athletes and non-athletes in general cognitive tasks (*Mallek et al., 2017*). Compared to the number of words used as prime and target stimuli in previous studies (*Martens, Ansorge & Kiefer, 2011*; *Ulrich, Adams & Kiefer, 2014*), the number of geometric forms used in the present study was small, and each geometric shape had a unique physical feature. Participants may have stored the geometric forms and their corresponding reactions in their working memory, subsequently forming a stimulus-reaction coupling (*Kiesel et al., 2006*; *Wang, Huo & Wang, 2012*). The present results indicated that the performance of athletes was significantly influenced by

the presentation of the geometric forms despite perception being unconscious, whereas the performance of non-athletes was not affected by the presence of geometric forms whatsoever. Athletes established an unconscious association between the stimuli and motor responses, whereas the non-athletes did not. The motor reactions of the athletes were likely facilitated or disrupted by the unconsciously perceived primes, with the unconsciously perceived geometric shapes activating the previously established stimulus-response assignment and triggering a motor response for situations of prime-target congruency. The response to the target would have been facilitated as a result of being pre-activated by the prime. By contrast, when the prime and target stimuli required a different response (i.e., were incongruent), the response to the target would have been interfered because the pre-activated response was inconsistent with the required target response. Athletes would have needed to timely inhibit the incorrect motor response tendency during incongruent trial types. Hence, athletes responded faster and committed fewer errors on congruent trials than on incongruent trials. In the present study, athletes seemed to learn the rules faster and more effectively than non-athletes, permitting the establishment of a more stable unconscious connection between the stimuli and motor response. By contrast, this connection was not stable in non-athletes and did not reach an automatic level. Thus, the stimuli presented below the threshold of consciousness didn't influence the performance of non-athletes. Although there was no unconscious response priming effect for non-athletes, error rates showed that non-athletes committed more errors on incongruent trials than on congruent trials, which paralleled a similar result with athletes. An aposteriori sample size based on the effect sizes of error rates was calculated in Gpower3.1, with the result showing that a minimum of 80 participants was needed with $\alpha = 0.05$, power $= 0.80$, effect size $= 0.32$. In order to find an equivalent main effect of response congruency, more non-athletes would be needed in future studies. We also found that the error rates among athletes were higher than those among non-athletes on incongruent trials, suggesting that stimulus-response conflict that was unconsciously perceived was stronger than non-athletes.

Moreover, our results showed that non-athletes did not exhibit a significant unconscious response priming effect. However, this finding was inconsistent with previous studies (*Klotz & Neumann, 1999*; *Neumann & Klotz, 1994*; *Ulrich & Kiefer, 2016*). For instance, the study of *Klotz & Neumann (1999)*, which tested geometric stimuli on normal subjects, found that the participants exhibited significant unconscious response priming effect, suggesting that motor activation could be trigged without conscious discrimination. We speculated that the reason for the inconsistent results between us and *Klotz & Neumann (1999)* might be the level of task difficulty. Only two geometric forms were used in the study of *Klotz & Neumann (1999)*, resulting in fewer the possible prime-target combinations compared to our current study. Participants in their study made a response according to the spatial position of the target (e.g., if the target on the right, participants made a response by pressing the right button). Thus, we believed that the localization task in the present study was more difficult than the study of *Klotz & Neumann (1999)*. *Kibele (2001)* and *Kibele (2006)* suggested that unconscious information processing may be obstructed in complex tasks. Along a similar vein, in studying the neural signature of the unconscious

response priming effect, *Ulrich & Kiefer (2016)* recruited college students to participant in a masked priming task that also used geometric forms. They found that the difference in RTs between congruent and incongruent trials was significant, with the RTs for congruent trials averaging 575 ms and the RTs for incongruent trials averaging 588 ms. In contrast, the difference was not significant in ER because of slow decisions. In the present study, the RT range is 500–501 ms, indicating that non-athletes focused on the current task with their rapid responses not compromising accuracy. So, it is possible that non-athletes would have been able to show an unconscious response priming effect provided they made slower responses. In turn, perhaps this would produce results, comparable to *Ulrich & Kiefer (2016)*. Although participants were normal subjects in both studies, the reaction strategies of these two groups might be different. Based on the previous researches (*Braver, Cole & Yarkoni, 2010*; *Capa & Bouquet, 2018*; *Van Gaal et al., 2011*), we speculated that the group difference might be responsible for the inconsistent results. Future studies could systematically explore the effect of individual differences or group differences on unconscious information processing both in the general and sport-specific domain.

Our findings, together with the results from previous studies, not only add further evidence to support that expertise is an important determinant of unconscious processing but also provide preliminary evidence that, compared with non-athletes, athletes exhibit superior unconscious information processing in the general domain. Thus, the influence of motor expertise on unconscious information processing may be transferable from the sports-specific domain to the general cognitive domain.

## LIMITATIONS

The present study had some limitations. Firstly, participants may have judged it too difficult to use full cognitive effort in consciously perceiving the prime. Thus, subjective threshold measurements such as the 5-point Likert scale should be considered in future studies, as it would contribute to a more enhanced data precision in comparison to the binary interview procedure in the prime identification task. Secondly, the athletes played table tennis, which is an open skills sport. In order to better identify the facilitation of motor expertise on unconscious information processing in the general cognitive domain, athletes who play closed skills sports should also be recruited in future studies. Thirdly, although the present study described the influence of motor expertise on the unconscious executive control of the general domain on a behavior level, the neural signature of this phenomenon remains unclear and warrants future exploration.

## CONCLUSIONS

The findings of our study provide evidence that motor expertise facilitates the unconscious processing of geometric forms, indicating that the influence of motor expertise on unconscious executive control is not just sports-specific but can be transferred to the general cognitive domain.

### Funding
The present study was funded by the National Natural Science Foundation of China (No. 31971023). The funders had no role in study design, data collection and analysis, decision to publish, or preparation of the manuscript.

### Grant Disclosures
The following grant information was disclosed by the authors:
National Natural Science Foundation of China: 31971023.

### Competing Interests
The authors declare there are no competing interests.

### Author Contributions
- Jiaxian Geng performed the experiments, analyzed the data, prepared figures and/or tables, authored or reviewed drafts of the paper, and approved the final draft.
- Fanying Meng conceived and designed the experiments, performed the experiments, analyzed the data, prepared figures and/or tables, authored or reviewed drafts of the paper, and approved the final draft.
- Chao Wang and Hanna Haponenko analyzed the data, authored or reviewed drafts of the paper, and approved the final draft.
- Anmin Li conceived and designed the experiments, performed the experiments, prepared figures and/or tables, and approved the final draft.

### Human Ethics
The following information was supplied relating to ethical approvals (i.e., approving body and any reference numbers):
This study received approval from the Ethics Committee of Shanghai University of Sport (No. 2018025).

### Data Availability
The raw data are available in the Supplementary Files.

### Supplemental Information
Supplemental information for this article can be found online at http://dx.doi.org/10.7717/peerj.9520#supplemental-information.

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
