# Peer review of "Motor expertise affects the unconscious processing of geometric forms"

_PeerJ, doi:10.7717/peerj.9520_

## Round 0.1 · original submission · Major Revisions

The two reviews below were provided by experts in the field. Both reviewers comment on the high quality of the presentation of the manuscript overall, however identify a number of issues that need to be addressed. My own reading of the manuscript converges with their concerns. In your revision, please address each of the comments raised, by either including additional information, revising the terminology used, or by acknowledging limitations of the design and alternative possible explanations of findings.

I urge you to take particular care to address the issues pertaining:

- representing priming effects as driven by executive control (reviewer 1)
- the distinctiveness of the current study (reviewer 1)
- the scope of the background literature (reviewer 2)
- sample size (reviewer 2)

Given that each of the issues raised by the reviewers can be addressed in a revision, I would like you to provide a revision of the manuscript, accompanied by a detailed letter describing your responses to each of the issues that were raised by the reviewers.

Reviewer 1 ·

Basic reporting

/

Experimental design

/

Validity of the findings

/

Additional comments

Dear Editor,
Geng et al. present a study in which they compare performance of athletes versus non-athletes on a masked priming task. Their findings show that only athletes show a significant masked priming effect, despite both groups displaying chance level performance when asked to identify the primes. The findings themselves seem convincing and the manuscript is well written, but I have major conceptual concerns. Most specifically i) these findings do not seem to speak about executive control, and ii) it is unclear how they differ from previous work.

Major comments
Throughout the manuscript, the authors consistently describe their effects as evidence for ‘superior unconscious executive control’. I think this is problematic for two reasons: i) it is unclear why showing a masked priming effect is ‘superior’, as the optimal thing to do in this task would be to ignore the primes. ii) a priming effect by itself cannot be used as a marker of executive control. Historically, priming effects are considered markers of bottom-up processing and not of top-down executive control. At line 235 the authors write “motor expertise possessed by athletes facilitates unconscious information processing” which seems like a fair and correct conclusion to me.

The difference between the current study and previous work is unclear to me. The authors argue that previous work has shown sport-specific priming effects, but e.g., Meng et al. used circles with notches in specific locations, which to me does not seem very sport-specific. A much better explanation should be given as to why that study was sport-specific and the current one isn’t.

The prime identification task is suboptimal because it is prone to a well-known task-difficulty artifact (cf Pratte & Rouder, 2009, AP&P). Because the identification task is much more difficult than the main task, participants might lose motivation, stop trying, and simply start pressing random buttons. Indeed, Pratte and Rouder (2009) showed that d’ significantly increases as clearly visible prime trials are intermixed with the prime identification task. Another classical problem is that prime identification tasks usually comprise of less trials and therefore have less power to reject the null hypothesis. I was unable to evaluate this because the number of trials used in the prime identification task was not mentioned.

Minor comments
The authors spend quite some space to argue that the lack of a priming effect in non-athletes compared to Kiefer et al. might be because their participants had less training trials. This seems like a very unlikely possibility (given that the difference was only 24 vs 32 trials), and more importantly, can’t this hypothesis easily be tested by excluding the first twelve trials from data analysis?

Frequentist statistics cannot provide evidence for the null hypothesis that primes were invisible, so these analyses should be corroborated using Bayesian statistics.

Please report whether or not the experimenters were aware of the hypothesis, and which groups participants where in. There is some evidence that priming effects can be explained by experimenter bias (cf. Doyen et al. 2012, PLoS ONE).

Line 242: the word intuition seems ill-chosen here. Probably it would be better to say that S-R mappings are so highly over trained in athletes that these can be used without any conscious deliberation (or something alike). This explains the data as well, without the vague notion of ‘intuition’.

Line 271-273, the use of the word ‘sense’ and ‘felt’ seems to implies that this was a conscious experience (cf. Desender et al., 2014, Psychol Sci). Please reword in order to avoid confusion.

Line 302: if the authors genuinely think their limited sample size is an issue, then why try to publish these data in the first place?

·

Basic reporting

The paper is written in clear and easily accessible English throughout the manuscript. The structure of the paper appears to conform to PeerJ standards. The figures of the manuscript appear to provide high-quality. They are sufficiently labelled and described and are relevant for the text. The research question is well-defined, relevant, and meaningful. The authors are successful in identifying a research deficit.

On a general scale, the issue of the paper relates to the processing of information in sports beyond a level of awareness of the very stimuli that elicit a motor response. Therefore, the PeerJ section Brain and Cognition provides a suitable platform for the publication of the results. The specific specific goal of the study relates to a previous study (Meng et al., 2019) ) of that author group in which they show that competitive table tennis players perceived table tennis specific visuospatial information on a computer screen without stimulus awareness whereas college students with no table tennis background did not. The present study extends these result as table tennis players and novices are tested in a similar set-up but with geometric symbols rather than table tennis specific stimuli. It is the goal of the authors to show that table tennis players are trained to process visual stimuli without awareness in general and not for visual stimuli representing the table tennis domain only.

While the purpose of this investigation can be readily followed it appears that the included literature is too narrow in scope while lacking a broader background on related psychological theories (Primary concern No. 1). In particular, the authors are encouraged to include the theory of direct parameter specification as outlined by Neumann and Klotz (e.g. Neumann & Klotz, 1994; Klotz & Neumann, 1999) as similar experimental paradigms with prime identification tasks were used.

References:

Klotz, W., & Neumann, O. (1999). Motor sctivation without conscious discrimination in metacontrast masking. Journal of Experimental Psychology: Human Perception and Performance, 25, 4, 976-992.

Neumann, O., & Klotz, W. (1994). Motor responses to nonreportable, masked stimuli: Where is the limit of direct parameter specification? In: C. Umilta, & M. Moscovitch (Eds.): Attention and Performance 15: Conscious and nonconscious inform a- tion processing. MIT Press: Cambridge MA, 123-150

Meng, F., Li, A., You, Y., & Xie, C. (2019). Motor expertise modulates unconscious rather than conscious executive control. PeerJ, 7. Doi: 10.7717/peerj.6387

Experimental design

The investigation is performed to a high technical and ethical standard. However, the authors may have strengthened their argument by conducting the tasks from their previous study (Meng et al., 2019) with sport specific stimuli along with the tasks from their present study within the same subject group while using a multivariate statistical analysis (Secondary concern No. 1).

The general methodological approach including a masked priming task and a prime identification task was adopted from the study by Zovko & Kiefer (2013). In addition, they authors may wish to compare their approach with the metacontrast paradigm as outlined by Neumann and Klotz (e.g. Neumann & Klotz, 1994; Klotz & Neumann, 1999). In particular, using a 5-point Likert scale for the prime identification task may contribute to an enhanced data precision in the prime identification experiment.

In general, the authors are encouraged to clarify their description of the prime identification task (lines 144-152). This section should be written in greater detail to outline how subjective and “objective measures of conscious awareness” determine prime stimulus awareness. It appears that a paper by Tulving (1993) may help to render the terms “consciousness” and “stimulus awareness” more precisely (Secondary concern No. 2) and clarify whether /or how a verbal response and/ or an emphasis on an accurate rather a speedy motor responses may be used to detect stimulus awareness (see also the dissociation paradigm by Erdelyi, 1986).

Two other methodological issues deserve notice. Firstly, in the masked priming task, subjects are asked to respond as quickly and accurately as possible (line 121) leaving open whether both goals are compatible at the same time. This issue seems crucial in regard to the speed-accuracy trade-off as advocated by Fitt’s Law. Therefore, equivocal meanings in the subject instructions should be avoided or else explained about in detail (Secondary concern No. 3). The second methodological issue raised here refers to the justification of the sample size. Contrary to other studies (Neumann & Klotz, 1994; Klotz & Neumann; 1999; Zovko & Kiefer, 2013), the present study failed to detect a significant difference in reaction times for normal subjects/ non-athletes between the congruent and the incongruent stimulus conditions while responding to geometrical stimuli. The speculation of the authors about a high perceptual sensitivity of athletes only towards geometrical stimuli must be considered rather vague while further empirical evidence is amiss. An alternative explanation is mentioned in the manuscript (line 269) indicating a tendency in the non-athletes to respond similar to the athletes on congruent versus incongruent trials. Possibly, a larger sample size might have provided statistical significance (Primary concern No. 2). Therefore, the authors are encouraged to justify their sample size through a Power analysis as required in various scientific journals already. In particular, the authors are encouraged to use the effect sizes of related studies (e.g. Zovko & Kiefer, 2013) to estimate their anticipated effect size along with the alpha- and beta-errors in the Power analysis.


References:

Tulving, E. (1993). Varieties of consciousness and levels of awareness in memory. In: A.D. Baddeley & L. Weiskrantz (Eds.): Attention, selection, awareness, and control (p. 283-299). Oxford: Claredon Press.

Erdelyi, M.H. (1986). Experimental indeterminacies in the dissociation paradigm. Behavioral and Brain Sciences 9, 30-31

Zovko, M., & Kiefer, M. (2013). Do different perceptual task sets modulate electrophysiological correlates of masked visuomotor priming? Attention to shape and color put to the test. Psychophysiology, 50, 2, 149-157.

Validity of the findings

Provided the validity of the present findings, athletes from combat sports or game sports would respond faster and with fewer errors in congruent versus in congruent trials with geometric symbols as compared to non-athletes. In particular, their perceptual sensitivity towards geometric symbols would be enhanced through warm-up trials while such enhancement is amiss for non-athletes.
However, care should be taken when interpreting the results of the competitive tale tennis players and the non-players. While, the authors mention a tendency in the non-players to respond similar to the players a sample size issue arises. Possibly, a larger sample size may have provided statistical significance. Therefore, the authors are asked to provide table data including effect sizes and to conduct an aposteriori sample size calculation using a Power analysis tool (e.g. GPower : Faul, Erdfelder, Lang, & Buchner, 2007).

References:

Faul, F., Erdfelder, E., Lang, A.-G., & Buchner, A. (2007). G*Power 3: A flexible statistical power analysis program for the social, behavioral, and biomedical sciences. Behavior Research Methods, 39, 175-191.

Additional comments

Your paper is written in clear and easily accessible English throughout the manuscript. Your manuscript follows a well-comprehensible line of thought. The methodology is adopted from previous studies in the literature. However, the missing reaction time difference between congruent and incongruent trials in the non-athletes remain inconclusive while contradicting other studies on one hand and while leaving open an alternative explanation based on a larger sample size on the other hand.

Some primary and secondary concerns are listed in this review. For the primary concerns, an inclusion of related literature on the metacontrast effect and the underlying direct parameter specification (Neumann & Klotz, 1994; Klotz & Neumann; 1999) is suggested. Moreover, I suggest to provide a sample size justification through a Power analysis to rule out a possible statistical cause for the missing significance in the two-way repeated measures ANOVA.
For the secondary concerns, it would strengthen your line of evidence if sport specific stimuli and geometric stimuli would be tested to identify a priming effect in congruent and incongruent trials in the same athlete and non-athlete subject groups. In addition, a clarification of terms and meanings could be achieved through a more in-depth review of existing psychological literature on what is called in the manuscript conscious awareness of the prime stimulus. As a last point, the authors are encouraged to explain about a possible conflict within the subjects when being instructed to respond a quickly and as accurately as possible at the same time.

---

## Round 0.2 · Minor Revisions

I agree with both reviewers that the manuscript has improved considerably through the revisions that were made, and I appreciate the detailed response letter. Nevertheless, certain issues that can be addressed through minor revisions remain outstanding. Please ensure in your revision to address these remaining comments from the reviewers:

- Include the difficulty artifact concerning the priming identification task as a limitation of the current study (Reviewer 1)

- The power analysis which is now included appears to rest on the assumption of a .8 correlation between the within-subjects variables. Please justify the origins of this assumption and the basis of expecting a medium-sized effect for the overall ANOVA (Reviewer 1)

- Discuss the results of similar papers showing clear RT differences between congruent and incongruent trials in normal non-athletic subjects, and expand your Discussion of why non-athletic subjects in the current study did not exhibit any RT difference between congruent and incongruent trials. (Reviewer 2)

- Include the differences in the assessment of prime awareness between the current study and previous approaches in the limitations (Reviewer 2)
- Proofread the manuscript thoroughly to eliminate spelling errors (Reviewer 2)

Reviewer 1 ·

Basic reporting

no comment

Experimental design

no comment

Validity of the findings

no comment

Additional comments

The authors have done a good job on revising their work based on my previous comments. I would like to further discuss two of my previously raised points, but apart from that have no further comments.

- About the prime identification task: the "difficulty artifact" that Pratte & Rouder discuss stems from the fact that the main task is relatively easy (~85% correct) whereas the detection task was very hard (~50%), so participants might feel that in the detection task they are merely guessing and thus "give up". The authors responded to this that participants were told there was no time pressure, but that doesn't solve the issue at all. That being said, the data are what they are, but it would be good if the author mentioned this shortcoming and take it into account for future work.
- I thought the author's response about differences between the current and previous work was not very convincing. That being said, I leave it up to the Editor to decide how much strength to put on novelty concerning publication.
- I am not familiar with Gpower, but a minimum of 16 participants for an effect size of .24 seems very little. This obviously depends on design specifics, but e.g., for a one-sample t test one would need to test 140 participants to achieve sufficient power for such a small effect. The authors might want to reassure that they did this correctly.

·

Basic reporting

1. Referring to my first review, substantial improvements have been provided in the re-submission of the paper by Geng et al. “Motor expertise affects unconscious processing of geometric forms”. In fact, through their re-analysis of previous data the authors show that motor expertise in a sport-specific domain is affecting unconscious processing somewhat independent of the stimulus structure.

However, while the authors have additionally mentioned about the theory of direct parameter specification they have failed to discuss the results of the related papers (e.g. Neumann & Klotz, 1994; Klotz & Neumann, 1999). More specifically, similar to the authors, Neumann and Klotz have used a masking paradigm with geometric forms in their experiments, as well, while consistently showing clear RT differences between congruent and incongruent trials in normal non-athletic subjects. These results add to other studies (e.g., Ulrich & Kiefer, 2016) showing RT differences between congruent and incongruent prime-target combinations in normal subjects as well. So, rather than discussing why athletes show superior processing of non-consciously perceived stimuli the authors should have provided arguments why their non-athletic subjects did not exhibit any RT difference between congruent and incongruent trials in the face of considerable evidence in the literature showing the contrary. In my opinion, individual differences, as advocated in the resubmission, provide only limited to no substantial explanation about these inconsistent results.

As a minor point, several writing errors should be eliminated prior to the publication in PEERJ (e.g. line 41 (…higher than in non-athletes…), line 160/161 (……prima and target included the same geometric stimulus…., …included a different geometric stimulus…), line 206 (…to a general domain, we, firstly, selected…), line 219 (…but the exact content…), line 311 (…of the present study…), line 319 (…these capacities are not limited to… ), etc.). Hence, the authors are encouraged to thoroughly re-correct their text and eliminate all writing errors. Please examine, as well, whether the statement line 38 (No significant difference was detected between athletes and non-athletes …for congruent trials) as compared to Fig. 3.

Experimental design

2. The new contributions of the authors concerning their experimental design have substantially improved their manuscript. In particular, additional evaluations regarding the stimulus specificity of the unconscious response priming effect and their extended outline on the prime identification control task are much appreciated.

3. I understand that the authors could not retrospectively change their analysis of objective and subjective measures of prime awareness. However, they could have discussed their approach in comparison to other approaches in the literature as a limitation of their study.

4. The authors efforts to clarify to a possible speed-accuracy trade-off conflict are much appreciated. However, I am still wondering about an equivocal meaning of the authors' instructions at the time when the experiment was conducted.

5. Again, based on the authors' results, I have no doubt about the pronounced perceptual sensitivity of the table-tennis players to sport-specific stimuli and rather general geometric stimuli. However, the authors' argument regarding individual differences when aiming to explain the similarity of RT in congruent and incongruent trials is not convincing owing to considerable evidence in the literature showing opposite results. Moreover, if individual difference would contribute to the different results in non-athletes the same could be true for the results in the athletes as well.

6. The authors efforts to provide a sample size justification is much appreciated.

Validity of the findings

7. The authors efforts to provide an aposteriori sample size calculation is much appreciated.

Additional comments

The authors effort to improve the paper is much appreciated. While few open issues were not thoroughly discussed in great detail the overall quality of the manuscript was substantially improved. Nevertheless, some minor revisions are needed in regard to writing errors.

---

## Round 0.3 · Minor Revisions

Your revisions address the remaining queries raised by the reviewers. However, the text in the revised manuscript still contains several typographical and grammatical errors, particularly in the newly added sections. As PeerJ performs no language or copy-editing before publication, please address these issues so the manuscript will be ‘production ready’ at acceptance.

---

## Round 0.4 · accepted · Accept

Thank you for your thorough review of typographical and grammatical errors in the text. I am happy to now accept this manuscript for publication. Congratulations!